

# It's a matter of design—how pitfall trap design affects trap samples and possible predictions

Fabian A. Boetzl[1], Elena Ries[2], Gudrun Schneider[1] and Jochen Krauss[1]

[1] Department of Animal Ecology and Tropical Biology, Biocenter, University of Würzburg, Am Hubland, Würzburg, Germany
[2] Regierungspräsidium Karlsruhe, Referat 56—Naturschaftspflege, Karlsruhe, Germany

## ABSTRACT

**Background:** Pitfall traps are commonly used to assess ground dwelling arthropod communities. The effects of different pitfall trap designs on the trapping outcome are poorly investigated however they might affect conclusions drawn from pitfall trap data greatly.

**Methods:** We tested four pitfall trap types which have been used in previous studies for their effectiveness: a simple type, a faster exchangeable type with an extended plastic rim plate and two types with guidance barriers (V- and X-shaped). About 20 traps were active for 10 weeks and emptied biweekly resulting in 100 trap samples.

**Results:** Pitfall traps with guidance barriers were up to five times more effective than simple pitfall traps and trap samples resulted in more similar assemblage approximations. Pitfall traps with extended plastic rim plates did not only perform poorly but also resulted in distinct carabid assemblages with less individuals of small species and a larger variation.

**Discussion:** Due to the obvious trait filtering and resulting altered assemblages, we suggest not to use pitfall traps with extended plastic rim plates. In comprehensive biodiversity inventories, a smaller number of pitfall traps with guidance barriers and a larger number of spatial replicates is of advantage, while due to comparability reasons, the use of simple pitfall traps will be recommended in most other cases.

## INTRODUCTION

Proposed nearly a century ago, pitfall traps remain one of the most commonly applied sampling methods in ecological field studies and are widely used for the assessment of ground dwelling arthropod taxa which are of high importance in modern ecosystem functioning research (*Brown & Matthews, 2016*). Although the limitations of pitfall traps in respect to trait filtering and reflecting diversity and abundances in a habitat appropriately have been intensively discussed, the method is still the best standardized and comparable approach to study ground dwelling arthropods and due to comparative low handling time allows for sufficient replication (*Driscoll, 2010*; *Kotze et al., 2011*).

Corresponding author
Fabian A. Boetzl,
fabian.boetzl@uni-wuerzburg.de

Different features of pitfall trap designs have undergone review and research over the last decades in order to improve and standardize trap designs: colour of traps (*Buchholz et al., 2010*), the presence and colour of rain covers (*Buchholz & Hannig, 2009*; *Császár et al., 2018*), sampling intervals (*Schirmel et al., 2010*), spatial distribution (*Ward, New & Yen, 2001*), different preservatives (*Schmidt et al., 2006*; *Skvarla, Larson & Dowling, 2014*) as well as pitfall trap diameters and the use of funnels (*Császár et al., 2018*; *Lange, Gossner & Weisser, 2011*). A recent meta-analysis by *Brown & Matthews (2016)* discussed many pitfall trap parameters (diameter, depth, colour, rain covers, preservatives and the use of funnels) and even proposed a standardized trap design. However, the authors did not consider additions to pitfall trap designs such as extended rim plates or guidance barriers although these have been used in previous studies.

In a conventional simple pitfall trap as proposed by the pioneers *Dahl (1896)*, *Barber (1931)* and *Greenslade (1964)* which is basically a container sunk in the soil, a large proportion of ground dwelling beetle species occurring in a habitat will not be detected with sufficient certainty as they are comparatively rare in assemblages (*Driscoll, 2010*). To overcome this limitation without increasing the sampling effort and workload, guidance barrier pitfall traps have been introduced. The use of guidance barriers is meant to increase capture efficiency and the number of singletons which are particularly important in conservation studies where complete biodiversity inventories are desired (*Hansen & New, 2005*). However, only few studies investigated the actual effect of guidance barriers on pitfall trap catches and these studies suffered from low sample sizes and unbalanced sampling efforts with a different number of pitfall traps used in the studied pitfall trap designs (*Hansen & New, 2005*; *Winder et al., 2001*). Moreover, guidance barrier pitfall traps are not commonly used in applied research (but see *Hossain et al. (2002)* and *Schneider et al. (2016)*). To our knowledge, the effects of barriers on assemblages or traits have so far not been investigated at all.

To reduce handling time in the field as well as the so called 'digging in effects' (*Digweed et al., 1995*; *Greenslade, 1973*), pitfall traps with extended polyvinyl chloride (PVC) rim plates screwed on top of a pitfall trap glass jar have been designed. With this design, hardly any digging or filling of gaps between the pitfall trap and the surrounding soil is needed in the process of exchanging pitfall trap containers and therefore handling time and the release of soil $CO_2$ are reduced. The latter is known to increase ground dwelling arthropod activity and could therefore affect pitfall trap catches and lead to an overestimation of activity densities (*Joosse & Kapteijn, 1968*; *Schirmel et al., 2010*). Another advantage of this design is the enhanced standardization of the transition between the pitfall trap and surrounding soil. However, it is unknown how these traps perform in comparison with simple pitfall traps and whether the extended plastic rim plate affects resulting ground dwelling arthropod assemblages.

In this study, we investigate the effects of different pitfall trap designs on the catches of the three most commonly studied ground dwelling arthropod taxa (carabid beetles, staphylinid beetles and spiders) and possible effects on assemblage structure and trait filtering in carabid beetles.

## MATERIALS AND METHODS

### Study design

The study was performed on a semi-natural meadow with hedgerows near the Biocenter of the University of Würzburg in central Germany (49.779416°N, 9.973360°E). On this meadow five plots of each four pitfall traps were established. The five plots always consisted of four different pitfall trap designs:

  i. Conventional simple pitfall traps (Fig. 1A);
 ii. Pitfall traps with a squared extended PVC rim plate screwed on top of the glass jar used as pitfall trap for avoidance of digging in effects and easier exchange ($10 \times 10$ cm$^2$; diameter 7 cm; Fig. 1B);
iii. V-shaped guidance barrier pitfall traps using two 75 cm long metal barriers (height: 7 cm, angle: 90°, proposed by *Smith (1976)*; Fig. 1C);
 iv. X-shaped guidance barrier pitfall traps using four 75 cm long metal barriers (height: 7 cm angle: 90°, proposed by *Morrill, Lester & Wrona (1990)*; Fig. 1D).

The minimum distance between plots was 45 m while the pitfall traps within a plot were placed in a row and separated from each other by 10 m to minimize possible detrimental effects of neighbouring pitfall traps on pitfall trap catches. All pitfall traps were located 2 m apart from an adjacent hedgerow to make trap samples comparable.

### Data collection

Pitfall traps were set up at the 13th March 2012 and emptied biweekly until the 22nd May 2012 which results in 100 trap samples (five plots × four pitfall traps per plot × five sampling intervals). As pitfall trap containers, we used conventional glass honey jars (height: 9 cm, diameter: 7.5 cm, transparent) filled with 200 ml three parts water with one part ethylene glycol (automobile antifreeze; H. Kerndl GmbH, Vaterstetten, Germany) mixture with odourless detergent as trapping liquid and preservant in each pitfall trap design. Pitfall traps were covered by metal roofs ($25 \times 25$ cm$^2$, approximately 10 cm above ground) to protect against flooding. To minimize small vertebrate bycatch, all traps except for the X-shaped guidance barrier type were additionally covered by a wire cross (length: 10 cm, width: 2 mm; Fig. 1B).

Pitfall trap samples were sorted for carabid beetles (Carabidae), rove beetles (Staphylinidae) and spiders (Araneae). All individuals from these taxa were counted and all carabid beetles were identified to species level following *Müller-Motzfeld (2006)*. Carabid body size was obtained from the online database 'carabids.org' (*Homburg et al., 2014*).

### Statistical analyses

All statistical analyses were performed in R 3.3.1 for Windows (*R Development Core Team, 2016*).

For all analyses the pitfall trap samples of the same plot and pitfall trap design were pooled over all five sampling intervals to account for phenological shifts in the assemblages and reduce the influence of outliers (*Kotze et al., 2011*; *Williams et al., 2010*).

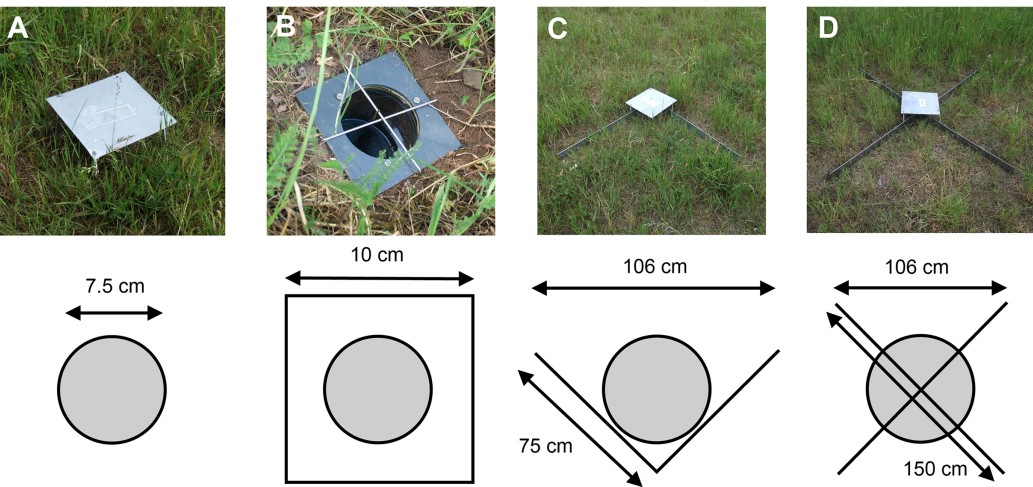

**Figure 1 Pitfall trap designs investigated.** (A) Conventional pitfall trap, (B) pitfall trap with a squared PVC rim (shown without metal roof), (C) pitfall trap with V-shaped guidance barriers, (D) pitfall trap with X-shaped guidance barriers. Arrows indicate measurements of the corresponding parts.

We used linear mixed effects models (LMER; using 'lmer' from the package lme4 (*Bates et al., 2015*)) to analyse the effects of pitfall trap type on the response variables 'carabid species richness,' 'carabid beetle activity density,' 'staphylinid beetle activity density,' 'spider activity density,' the 'proportion of small (<6 mm) carabid beetles,' the 'proportion of large (>10 mm) carabid beetles' and 'carabid beetle community weighted mean body size.' Activity densities were square root transformed to account for heteroscedasticity. All LMERs contained 'plot ID' as a random effect to account for the nested design and were analysed using the command 'anova' (type II sums of squares, Kenward–Roger approximation of denominator degrees of freedom) from the package 'lmerTest' (*Kuznetsova, Brockhoff & Christensen, 2017*) and subsequent Tukey-HSD post hoc tests ('HSD.test' command from the package agricolae (*De Mendiburu, 2016*)). All models met the model assumptions for linear mixed effects models. All response variables were tested for spatial autocorrelation using the 'Moran.I' command from the package ape (*Paradis, Claude & Strimmer, 2004*) but no spatial autocorrelation was detected.

Additionally, we compared the resulting carabid assemblages (transformed to proportion data) in a non-metric multidimensional scaling (NMDS) based on Bray–Curtis dissimilarities using the 'metaMDS' command (package 'vegan' (*Oksanen et al., 2013*)). The NMDS fit was tested for significant differences between pitfall trap designs in a PERMANOVA ('adonis' command). To compare the capture efficiency of the different pitfall trap designs we calculated species accumulation curves using the 'specaccum' function (including 95% confidence intervals).

## RESULTS

On the five plots, a total of 655 carabid beetles from 42 species, as well as 626 staphylinid beetles and 4,557 spiders were trapped over the course of the five sampling intervals (Tables 1 and 2).

**Table 1 Total activity densities and carabid species richness in the different pitfall trap designs over all plots and sampling intervals.**

| Response variable | (i) Simple type | (ii) Extended PVC rim plate type | (iii) V-shaped barrier type | (iv) X-shaped barrier type |
|---|---|---|---|---|
| *Carabid beetles* | | | | |
| Species richness | 17 | 17 | 29 | 35 |
| Activity density | 55 | 39 | 208 | 353 |
| *Staphylinid beetles* | | | | |
| Activity density | 103 | 44 | 131 | 348 |
| *Spiders* | | | | |
| Activity density | 576 | 354 | 973 | 2,654 |

Activity densities of all three investigated taxa were at least three times higher in the pitfall traps with X-shaped guidance barriers compared to simple pitfall traps and lowest in the pitfall traps with extended PVC rim plate (not significant in carabid beetles; Table 1; Fig. 2). The pitfall traps with V-shaped guidance barriers caught more carabid beetles and spiders than the simple ones but less than the pitfall traps with X-shaped guidance barriers in all three taxa (carabid beetles: LMER: $F_{3,12} = 69.0$, $p < 0.001$; Fig. 2A; staphylinid beetles: LMER: $F_{3,12} = 35.8$, $p < 0.001$; Fig. 2B; spiders: LMER: $F_{3,12} = 99.7$, $p < 0.001$; Fig. 2C).

The carabid assemblages in the pitfall trap samples of the pitfall trap designs using guidance barriers had approximately twice as many species than the assemblages in the two remainder types (LMER: $F_{3,12} = 20.6$, $p < 0.001$; Fig. 3A). The pitfall traps with extended PVC rim plate caught significantly less small species (<6 mm; LMER: $F_{3,12} = 12.3$, $p < 0.001$) and more large species (>10 mm; LMER: $F_{3,12} = 4.38$, $p = 0.027$) which resulted in a larger community weighted mean body size than in the other pitfall trap designs (LMER: $F_{3,12} = 69.0$, $p < 0.001$; Fig. 3B).

The NMDS ordination (stress 0.18) showed considerably less variation among the very similar carabid assemblages in the pitfall traps with guidance barriers while carabid assemblages in the pitfall trap designs without guidance barriers showed larger variation between plots. Pitfall trap design influenced the trapped assemblages significantly (PERMANOVA: 9,999 permutations, $F_{3,19} = 2,68$, $p < 0.001$) as the pitfall traps with extended PVC rim plates had rather different assemblages compared to the other pitfall trap types (Fig. 4A).

The species accumulation curves indicated that using one single pitfall trap with X-shaped guidance barriers is approximately as effective as using four to five simple pitfall traps (Fig. 4B).

# DISCUSSION

We showed that the choice of pitfall trap design affects the final outcome of a study and results in different assemblage approximations even though we tested one habitat which has a finite carabid species pool and a unique assemblage structure. Pitfall trap design might therefore affect conclusions drawn from the study outcome and pitfall trap designs unsuitable for the hypotheses will result in poor results and poor management recommendations.

**Table 2 Carabid beetle species present in the different pitfall trap designs over all plots.**

| Species | (i) Simple type | (ii) Extended PVC rim plate type | (iii) V-shaped barrier type | (iv) X-shaped barrier type |
|---|---|---|---|---|
| *Abax parallelepipedus* | | | | 1 |
| *Amara aenea* | 1 | 1 | 14 | 30 |
| *Amara convexior* | 3 | 2 | 4 | 17 |
| *Amara equestris* | | 1 | | |
| *Amara eurynota* | | | 1 | 1 |
| *Amara familiaris* | 1 | | 3 | 3 |
| *Amara lucida* | | | 1 | 3 |
| *Amara lunicollis* | 1 | | 1 | 2 |
| *Amara ovata* | | 1 | 1 | 5 |
| *Amara similata* | | | 3 | |
| *Anchomenus dorsalis* | | | 1 | 4 |
| *Badister bullatus* | 1 | 1 | 1 | 1 |
| *Badister lacertosus* | | | | 2 |
| *Bembidion lampros* | 2 | 1 | 5 | 13 |
| *Bembidion properans* | | | | 2 |
| *Brachinus crepitans* | | | 1 | 6 |
| *Calathus melanocephalus* | 2 | | 3 | 2 |
| *Carabus coriaceus* | 1 | 1 | | |
| *Harpalus affinis* | | 1 | | 1 |
| *Harpalus pumilus* | 3 | 1 | 20 | 32 |
| *Harpalus rubripes* | 5 | 9 | 21 | 62 |
| *Harpalus rufipes* | 1 | 1 | 1 | |
| *Harpalus subcylindricus* | | 2 | 2 | 18 |
| *Harpalus tardus* | 8 | 9 | 27 | 54 |
| *Lebia cruxminor* | 1 | | | |
| *Masoreus wetterhalii* | | | 2 | |
| *Microlestes maurus* | | 1 | 2 | 1 |
| *Notiophilus biguttatus* | | | | 1 |
| *Notiophilus palustris* | | | 2 | 2 |
| *Ophonus laticollis* | | | 1 | |
| *Panagaeus bipustulatus* | 1 | | 2 | 3 |
| *Paradromius linearis* | | | 1 | 2 |
| *Philorhizus notatus* | | | 2 | 1 |
| *Philorhizus sigma* | | | | 1 |
| *Poecilus cupreus* | 4 | 1 | 10 | 11 |
| *Poecilus lepidus* | | 5 | 10 | 25 |
| *Poecilus versicolor* | | | | 1 |
| *Pterostichus strenuus* | | | | 5 |
| *Syntomus foveatus* | | | 7 | 2 |
| *Syntomus truncatellus* | 19 | 1 | 59 | 37 |
| *Synuchus vivalis* | | | | 1 |
| *Trechus quadristriatus* | | | | 1 |

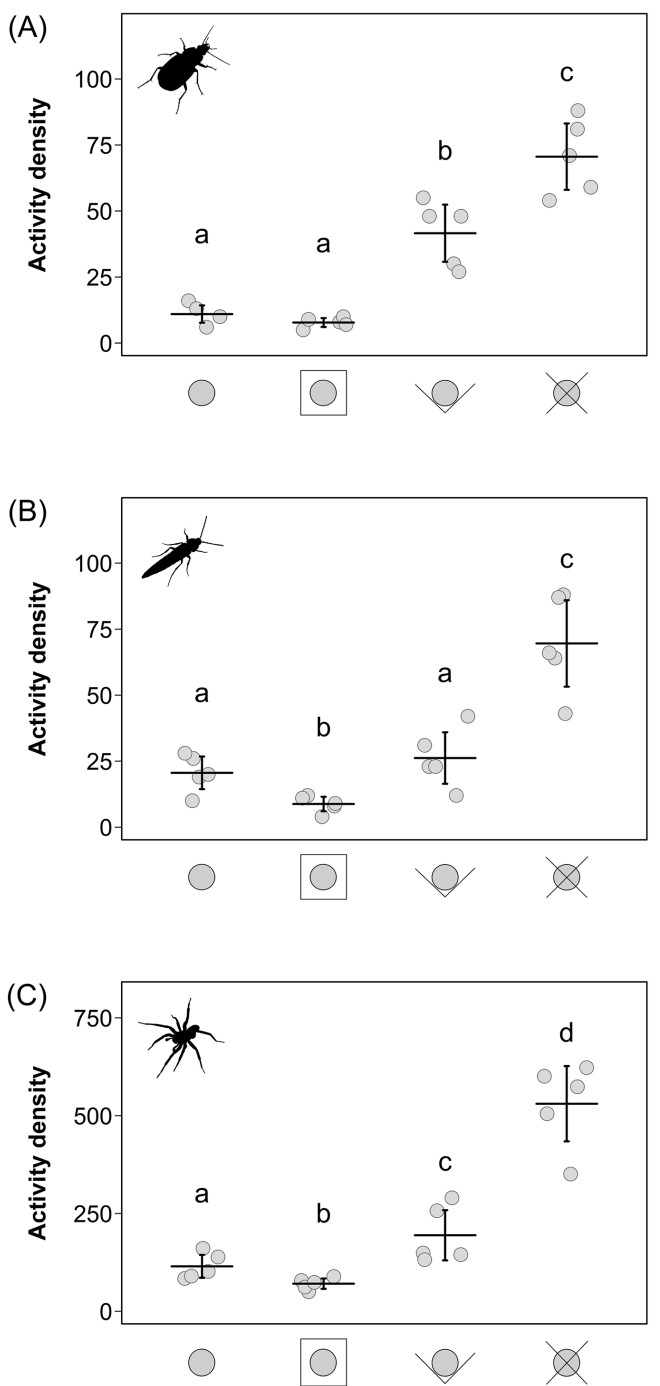

**Figure 2 Activity densities (mean ± 95% CI) of carabid beetles (A), staphylinid beetles (B) and spiders (C) for the four different pitfall trap types.** Different letters above indicate significant differences ($p < 0.05$). Pitfall trap designs from left to right: simple type; extended PVC rim plate type; V-shaped guidance barrier type; X-shaped guidance barrier type.

Pitfall traps with guidance barriers proved most effective in our study and showed coherent carabid beetle assemblage structures. These pitfall trap designs have been proposed several times in the past but have seen little use so far (*Durkis & Reeves, 1982*;

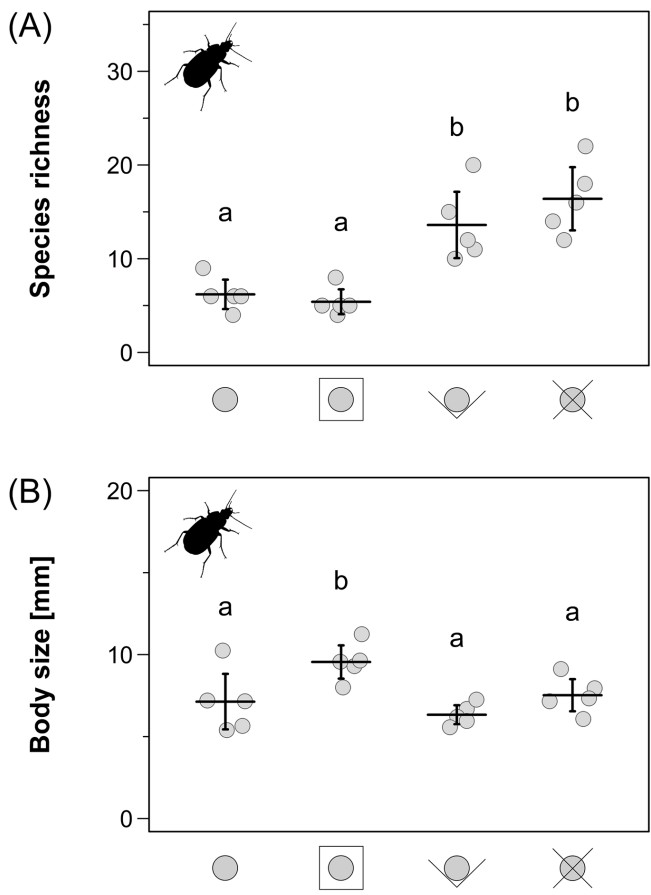

**Figure 3 Carabid beetle species richness (A) and community weighted mean body size within the carabid assemblages (B) for the four different pitfall trap types.** Means ± 95% CI. Different letters above indicate significant differences ($p < 0.05$). Pitfall trap designs from left to right: simple type; extended PVC rim plate type; V-shaped guidance barrier type; X-shaped guidance barrier type.

*Hansen & New, 2005*; *Winder et al., 2001*). According to our results, the use of a few pitfall traps with guidance barriers would be recommended over the use of many simple pitfall traps, especially when comprehensive biodiversity inventories are desired. One single pitfall trap with guidance barriers was approximately as effective as four to five simple pitfall traps in regards of species richness; therefore, the use of pitfall traps with guidance barriers could spare handling time in the field and in the laboratory. The guidance barrier designs are (i) more time consuming in the construction with higher payloads in the field and (ii) at least in the X-shaped design traps need to be renewed in each sampling interval in order to exchange the trap container (a problem that could probably be overcome by revisiting the design of the guidance barriers). Nonetheless, a reduced overall number of traps and overall reduced handling time subsequently saved in sample processing later on increases the efficiency. However, pitfall traps with guidance barriers need a more or less plain ground: for the study of uneven surfaces (e.g. rocky ground), simple pitfall traps will be more suitable and efficient. The increased visibility of guidance barrier traps could also result in a higher rate of vandalism which is a problem in pitfall

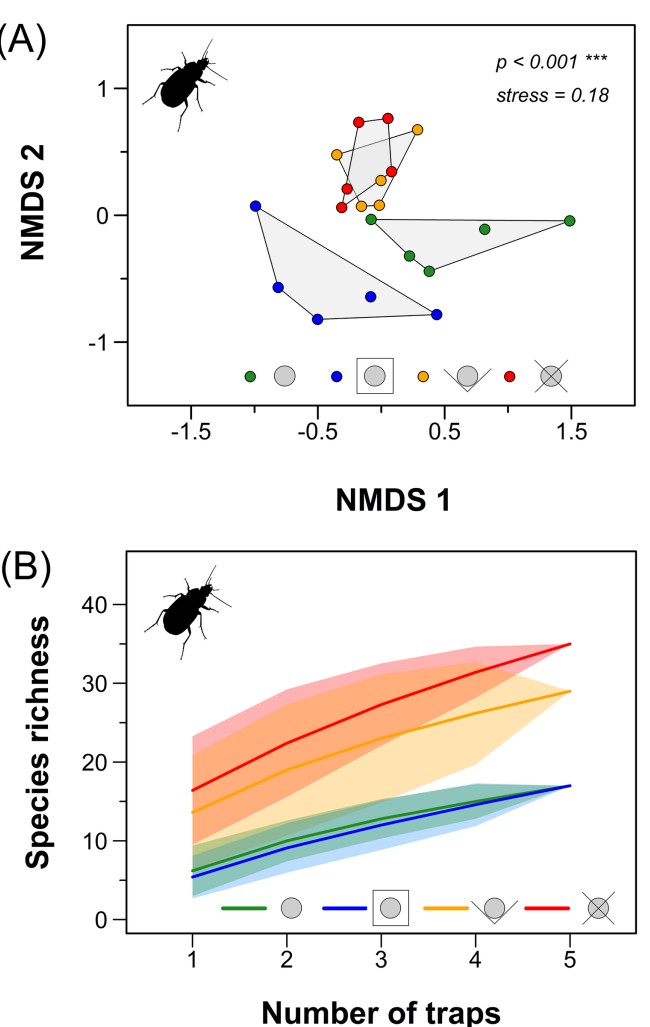

**Figure 4 Carabid beetle assemblage characteristics.** (A) NMDS ordination of the carabid assemblages in the different pitfall trap types. (B) Species accumulation curves (±95% CI) for the four different pitfall trap types. Colours: green: conventional pitfall trap; blue: pitfall trap with a squared PVC rim; orange: pitfall trap with a V-shaped guidance barrier; red: pitfall trap with a X-shaped guidance barrier. Pitfall trap designs from left to right: simple type (green); extended PVC rim plate type (blue); V-shaped guidance barrier type (orange); X-shaped guidance barrier type (red).

trap studies but might be minimized by using transparent barriers as used by *Durkis & Reeves (1982)*. Moreover, as for now not many studies used pitfall traps with guidance barriers, for means of comparability between studies it will be necessary to use simple pitfall traps in many fields of ecological research.

In contrast to *Winder et al. (2001)*, we did not find barrier pitfall trap designs to catch more carabid beetles of a certain size as community weighted mean body size and percentages of small and large carabid beetles remained approximately the same as in the simple pitfall traps. The shape of the guidance barriers only affected total activity densities and not species richness or assemblage composition, therefore, the V-shaped guidance barrier pitfall traps which have a lower impact on ecosystems as their catchment area is only a quarter of those of the X-shaped guidance barrier pitfall traps can be used in

biodiversity inventories. This design has also the advantage that the guidance barriers are not covering the pitfall trap container which makes the exchange or collection of pitfall trap samples much easier and faster.

Although simple pitfall traps caught less species and fewer individuals as the pitfall traps containing guidance barriers, the resulting carabid assemblages were over all rather similar to those from the pitfall traps with guidance barriers. The remaining differences could probably be compensated by using a higher number of simple pitfall traps in habitats and situations where the use of guidance barriers is not possible.

The use of extended PVC rim plates mainly aims at reducing set up time for pitfall traps and digging-in effects. Although this method is—apart from the use of funnels in combination with a two cup system as described in *Brown & Matthews (2016)*—probably the most efficient in terms of handling time in the field and quite good for standardizing the transition between surrounding soil an pitfall trap, it resulted in low catch rates and considerable alteration of the resulting carabid beetle assemblages in comparison to all other tested pitfall trap designs. Yet as activity densities were not drastically different from those in the simple pitfall traps, we argue that the 'digging in effect' did not boost activity densities essentially in our study.

However, trait filtering against small sized carabid species was apparent when using extended PVC rim plates. This effect is completely contrary to the original idea of a more standardized transition between surrounding soil and pitfall trap which should help to catch more small carabid beetles as pitfall trap sampling bias due to the perception of edges is decreasing with increasing body size (*Engel et al., 2017*; *Lang, 2000*). Carabid beetles most likely avoided the relatively smooth PVC surface and as smaller species would have to cover longer distances in relation to their body size, this effect results in trait filtering. It might be reduced by covering the surface with sand (fixed with spray glue) or other more natural organic substrates. For the moment, due to the reduced and size-biased catches produced by this pitfall trap design, we suggest to refrain from its use.

## CONCLUSIONS

Many studies on ground dwelling predators (carabid beetles, staphylinid beetles, spiders) use pitfall traps to draw conclusions on habitat conservation and ecosystem services including functional traits of species without being aware of the bias of certain pitfall trap designs. We show that pitfall trap design not only affects species richness and activity densities but also the trait composition of the resulting assemblages. For comprehensive biodiversity inventories, we therefore recommend to use a small number of highly effective pitfall traps with overall spared handling time, as the X- or V-shaped barrier trap designs in our study. However, for most ecological studies the use of simple pitfall trap designs is recommended, to increase comparability to other studies. We hereby recommend to use a nested design of several simple pitfall traps per plot followed by a pooling of the data as suggested by *Kotze et al. (2011)* in order to get a more conclusive picture of the local assemblages. We can currently not recommend the use of pitfall traps with an extended PVC rim plate, as especially small species tend not to cross such plates.

## ACKNOWLEDGEMENTS

We are grateful to Sascha Buchholz and two anonymous reviewers for their helpful comments on the manuscript and to the nature conservation authorities of Lower Frankonia for permits.

### Funding

Fabian A. Boetzl was supported by a grant of the German Excellence Initiative to the Graduate School of Life Sciences, University of Würzburg. This publication was funded by the German Research Foundation (DFG) and the University of Wuerzburg in the funding programme Open Access Publishing. The funders had no role in study design, data collection and analysis, decision to publish, or preparation of the manuscript.

### Grant Disclosures

The following grant information was disclosed by the authors:
German Excellence Initiative to the Graduate School of Life Sciences, University of Würzburg.
German Research Foundation (DFG).
University of Wuerzburg.

### Competing Interests

The authors declare that they have no competing interests.

### Author Contributions

- Fabian A. Boetzl analyzed the data, contributed reagents/materials/analysis tools, prepared figures and/or tables, authored or reviewed drafts of the paper, approved the final draft.
- Elena Ries conceived and designed the experiments, performed the experiments, analyzed the data, contributed reagents/materials/analysis tools, authored or reviewed drafts of the paper, approved the final draft.
- Gudrun Schneider conceived and designed the experiments, contributed reagents/materials/analysis tools, authored or reviewed drafts of the paper, approved the final draft.
- Jochen Krauss conceived and designed the experiments, contributed reagents/materials/analysis tools, authored or reviewed drafts of the paper, approved the final draft.

### Data Availability

   The raw data are provided in the Supplemental Files.

## Supplemental Information

Supplemental information for this article can be found online at http://dx.doi.org/10.7717/peerj.5078#supplemental-information.

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
