# Peer review of "It’s a matter of design—how pitfall trap design affects trap samples and possible predictions"

_PeerJ, doi:10.7717/peerj.5078_

## Round 0.1 · original submission · Minor Revisions

Dear Dr. Boetzl and colleagues:

Thanks for submitting your manuscript to PeerJ. I have now received three independent reviews of your work, and as you will see, the reviewers raised some minor concerns about the manuscript and research. In particular, please consider comments about experimental design, explanation of protocols and approaches, defining terms, accuracy in cited literature (accounting for missing references) and improvements to overall presentation (figures and also English). Therefore, I am recommending that you revise your manuscript accordingly, taking into account all of the issues raised by the reviewers. I do believe that your manuscript will be ready for publication once these issues are addressed.

Good luck with your revision,

-joe

·

Basic reporting

Thank your for this very interesting study on different pitfall trap designs and how these affect invertebrate samples. I really enjoyed reading as the article is concise, straight-forward and well written. The introduction is very good and all relevant references are cited. The question addressed in this study are very important and will improve our knowledge regarding the applicability of pitfall traps. The idea to test the performance of traps with barriers and to compare them to other designs is timely and very smart. I have some remarks (see below) that should be addressed before final acceptance.

Experimental design

The design is well-conceived and appropriate to answer the research questions. The methods are described with sufficient details but regarding the statistics I have three questions:
1. Why did you transform the carabid data to proportion data instead of using count data for NMDS? Please explain. I am concerned that this might obliterate differences in species composition. I was thinking about this because you found twice as much species in traps with barriers than in the other designs and I think that this should lead to a higher dissimilarity in species composition.
2. The fit was tested ... Which fit? What do you mean? Did you test if trap design affected the species composition by using the PERMANOVA? If yes and if test results were significant please be more careful in stating that differences in assemblage composition were rather low (see discussion).
3. I know that this means a lot of additional work but I would like to recommend to run analyses at least on spider species data. Maybe you could manage the identifications of spiders as well?

Validity of the findings

Data is robust - although I would be happy to see data on spider species as well. However, all conclusions are well stated, but two questions remain:
1. You recommend the use of pitfall traps with barriers but you should keep in mind that vandalism is a common problem in pitfall trap studies (especially in cities). Pitfall traps are better visible with barriers so I would expect more destruction.
2. Pitfall traps with barriers yielded significant higher species numbers. But at the same time carabid assemblages are rather similar among all pitfall trap designs. Please check the validity of the NMDS again and re-run the analysis with count data. If there are still no differences in species composition you should add a few sentences to your discussion. In particular, if there are no differences in species composition it may be not necessary to kill more individuals with barrier traps. But however, this clearly depends on the reserach questions. If one is interested in species inventories, using traps with barriers can be reasonable. In this context, it would be very interesting to analyse if the number of species of conservation concern (RL species) is also higher in barrier traps.

Additional comments

Thanks again for giving me the opportunity to read and review this very interesting study.

Reviewer 2 ·

Basic reporting

The basic reporting is fine except for some minor points listed under general comments.

Experimental design

The experimental design is fine regarding differences between single traps of various types. However, the alternative of using clusters of multiple simple versus single (or few) more sophisticated traps has not been directly investigated (as clusters of simple traps were not studied). Additional data would be needed, here, or more careful wording in parts of the discussion. Information on the handling time and cost of the different types of traps is also highly relevant and should be added to the manuscript if possible. See also general comments.

Validity of the findings

See comment on the limits of the experimental design.

Additional comments

Some minor comments and explanations of the above statements (in the order of appearance in the manuscript).

L27: Long sentence - better split into 2-3.

L31: As traditions differ, it is not clear what the conventional pitfall type is. Please use a more precise term. "simple" would already be better (as is already used in the discussion).

L34: ...more effective than...

L34: Please specify how much more effective p.t. with guidance barriers are (20% more, ten times as, etc.)

L38: I suggest to replace "detrimental effects" with something more direct, such as "reduced catches".

L63: Replace "regard" with "consider".

L66: "pit in the soil" is misleading - it should be clear that a container is involved, e.g. "container sunk in the soil"

L69: Additional traps placed in the same sampling location are usually not replicates with respect to the study design. Better use "sampling effort", here.

L71: Do singletons increase accuracy? I think accuracy applies to the common species that are caught in high numbers. Singletons are always highly stochastic.

L87: "detrimental effects on ... arthropod assemblages" implies that the local populations would suffer from the presence of these structures. You probably mean reduced number or different composition of the catch. This is already said in the first half of the sentence, so the second half can be removed.

L88: "capture efficiency" is return per effort, but you provide no information on effort (e.g. cost, or handling time of the different types compared). Better use a more neutral term such as "catches".

L163: replace "much" with "many"

L180, 183: pitfall trap studies can rarely count as experiments in a strict sense - better use "survey" or, neutral, "study".

L196: The question is (a) if multiple (e.g. 4) "conventional" traps together produce a more or less meaningful catch than a single X-barrier trap, and (b) if the effort (handling time and cost) are higher for either alternative. The lack of comparison between groups of "conventional" to single X-barrier traps does not allow strong conclusion in that respect. As an indication, you could combine the information of 4 (or 5) type A and type B traps in Fig. 4 to approximate the result obtained with such a trap cluster (you can only approximate, because your single A and B traps were spatially spread out over a much larger area than it would be done if they were used as clusters to characterize some local assemblage). I would be curious if the combined A type falls within the single X-barrier traps in terms of species composition.

L197: How do you know that the catch of a single X-barrier trap is not comparable to the sum of four A type traps? Your sentence also suggests that comparisons between studies are possible as long as A type traps are used. Given all the other sources of variation (material, cover, fluid, time, person), comparability is always limited.

L202: How reliable are the results of these species accumulation curves? There were less species in the V than in the X traps, the difference was just not significant. I agree that catching almost as many species among only a quarter of the individuals is striking. Nevertheless, a more careful wording may be advisable, here. Based on your results (and my guess that installation of V is nearly as laborious as X), I think there are good reasons to recommend X-shape (or multiple A) rather than V.

L219: How do you know that the higher catch in A than B traps are not due to the 'digging in effect'? What puzzles me, however, is why the 'digging in effect' is always seen as something negative on the one hand, and on the other hand people put effort in catching more individuals (with barriers etc.). If the 'digging in effect' boosts catches - fine!

L221: Replace "in this design" with "when using PVC rims".

L228: "...due to the reduced and size-biased catches produced by this pitfall..."

L233-235: Please write something more direct. The conclusion from your study is that the type of trap not only affects numbers of individuals and species caught, but also their trait composition.

L239: Please rewrite in a way in which readers get an idea what kind of a nested design you recommend without reading Kotze et al. (2011) in addition to your article.

Reviewer 3 ·

Basic reporting

See general comments to authors

Experimental design

See general comments to authors

Validity of the findings

See general comments to authors

Additional comments

This manuscript describes a comparative study with a focus on pitfall trap efficiency in cases where an ordinary trap is coupled with an expanded plastic rim plate or with two different guidance fence types. The authors compared the catches of carabid and rove beetles, and spiders, by setting three traps of each “design” in a meadow, and by analyzing the data using GLM, NMDS and abundance plots. The data as such are surprisingly large and the analysis methods are appropriate. Also, the paper is clearly written so my concerns may be considered relatively minor.

1. I understand that this is not a literature review but it is still striking that only three references are before the year 2000. Many key references are clearly missing: for example, the authors refer to “digging-in effect” without citing Digweed et al. (1995/Pedobiologia) or any of the Greenslade or Barber papers from the mid-1900s, let alone many others since then. I suggest the authors make a more thorough literature search and add these original references instead of rather frequent use of a few review papers from 2000s.
2. Summary may benefit from fine tuning. The first sentence is too long and should be split into two. Line 31 notes “rim” but any jar, cup or bottle has a rim: it may be better to call the structure “an extended rim plate” (or something like that). Line 32 mentions “guidance barriers” but strictly speaking these are not really barriers (that would hinder beetle movements) but rather “guiding fences” (or something like that). Regarding “rim” and “barrier”, I strongly suggest check the text throughout to clarify and fix these. Moreover, line 35 requires clarifications (not understandable currently).
3. Text, line 78 should add reference Digweed et al. 1995/Pedobiologia.
4. Please clarify the sampling design a bit: within each plot, how exactly were the traps placed relative to each other: in a row, in a square, randomly…? Also, In data collecting section perhaps say clearly that the trap as such was similar for each “design”.
5. The number of replicates per design was very low (3) which reduces the analysis power and makes the estimates of mean and variance difficult. Perhaps add a few words about this to Discussion.
6. It is kind-of good to check spatial autocorrelation but I would suspect that the use of plot as a random variable takes care of that (as it apparently indeed did).
7. Discussion is very thin as only four references were cited here! I strongly suggest do a thorough literature search and cite many more papers than currently (see comment #1).
8. Discussion could perhaps briefly note that the use of fences to increase catches might result in biases toward more actively moving species.
9. Please add totals for carabids, rove beetles and spiders to the different columns of Table 1.

---

## Round 0.2 · accepted · Accept

Dear Dr. Boetzl and colleagues:

Thanks for revising your manuscript based on the minor concerns raised by the reviewers. I now believe that your manuscript is suitable for publication. Congratulations! I look forward to seeing this work in print, and I anticipate it being an important resource for the Entomology, Ecology and Environmental Science fields. Thanks again for choosing PeerJ to publish such important work.

Best,

-joe

#